# Epidemiological impacts of the NHS COVID-19 app in England and Wales throughout its first year

Michelle Kendall [1] ✉, Daphne Tsallis[2], Chris Wymant [3,4], Andrea Di Francia[5], Yakubu Balogun[5], Xavier Didelot [1,6], Luca Ferretti[3,4] & Christophe Fraser [3,4,7]

The NHS COVID-19 app was launched in England and Wales in September 2020, with a Bluetooth-based contact tracing functionality designed to reduce transmission of SARS-CoV-2. We show that user engagement and the app's epidemiological impacts varied according to changing social and epidemic characteristics throughout the app's first year. We describe the interaction and complementarity of manual and digital contact tracing approaches. Results of our statistical analyses of anonymised, aggregated app data include that app users who were recently notified were more likely to test positive than app users who were not recently notified, by a factor that varied considerably over time. We estimate that the app's contact tracing function alone averted about 1 million cases (sensitivity analysis 450,000–1,400,000) during its first year, corresponding to 44,000 hospital cases (SA 20,000–60,000) and 9,600 deaths (SA 4600–13,000).

The NHS COVID-19 app was launched on 24 September 2020 across England and Wales, with millions of users installing it in the first few days after its launch. Its development was motivated by the theoretical finding that rapid, scalable and anonymised contact tracing could help reduce transmission of SARS-CoV-2[1–4]. It uses Google and Apple's Bluetooth exposure notification platform[5] and includes a range of services alongside digital contact tracing. These services have varied over time but include: reporting positive test results, local area information, venue check-in, symptom checking, test ordering, self-isolation countdown, links to public health advice, and access to self-isolation payments[6]. The system is privacy-preserving by design[7].

Evaluating the effectiveness of digital contact tracing interventions is important for their development, for policy making and for public trust[8]. An evaluation of the initial roll-out of NHS Test and Trace on the Isle of Wight, which included the first version of the NHS COVID-19 app, found a marked improvement in the course of the epidemic on

the island but there was no data available to separate out the impact of the app from the other aspects of the intervention[9]. An analysis of the epidemic in England and Wales from 24 September 2020 to 31 December 2020, which included the beginning of the surge in infections caused by the Alpha variant, estimated that during its first three months the NHS COVID-19 app reduced the total number of cases by 13% (central 95% range of sensitivity analyses 5–19%) or 24% (95% confidence interval 14–33%) depending on attribution technique[10]. Though the overall study was observational, there was an unintended experimental element that is useful for causal attribution: after review, the app settings were changed in the release of Version 3.9 on 29 October 2020, resulting in 2.9 times more contact tracing notifications per index case, and the estimated prevention effect similarly increased by a factor of 2.4–2.8[10].

Here we describe and evaluate the operation and impact of the NHS COVID-19 app over its whole first year. We present the first

[1]Department of Statistics, University of Warwick, Coventry CV4 7AL, UK. [2]Zühlke Engineering Ltd, 80 Great Eastern St, London EC2A 3JL, UK. [3]Big Data Institute, Li Ka Shing Centre for Health Information and Discovery, Nuffield Department of Medicine, Old Road Campus, University of Oxford, Oxford OX3 7LF, UK. [4]Pandemic Sciences Institute, Nuffield Department for Medicine, University of Oxford, Old Road Campus, Oxford OX3 7DQ, UK. [5]UK Health Security Agency, Nobel House, 17 Smith Square, London SW1P 3JR, UK. [6]School of Life Sciences, University of Warwick, Coventry CV4 7AL, UK. [7]Wellcome Centre for Human Genetics, Nuffield Department of Medicine, University of Oxford, Roosevelt Drive, Headington, Oxford OX3 7BN, UK. ✉e-mail: michelle.kendall@warwick.ac.uk

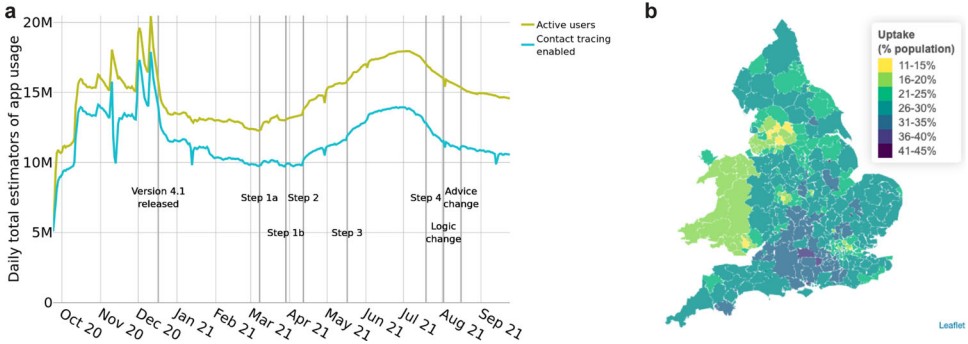

**Fig. 1 | App usage. a** The number of active app users across England and Wales, and the number of devices with Bluetooth contact tracing enabled. **b** App uptake per LTLA, estimated as the mean number of active users as a proportion of the total population.

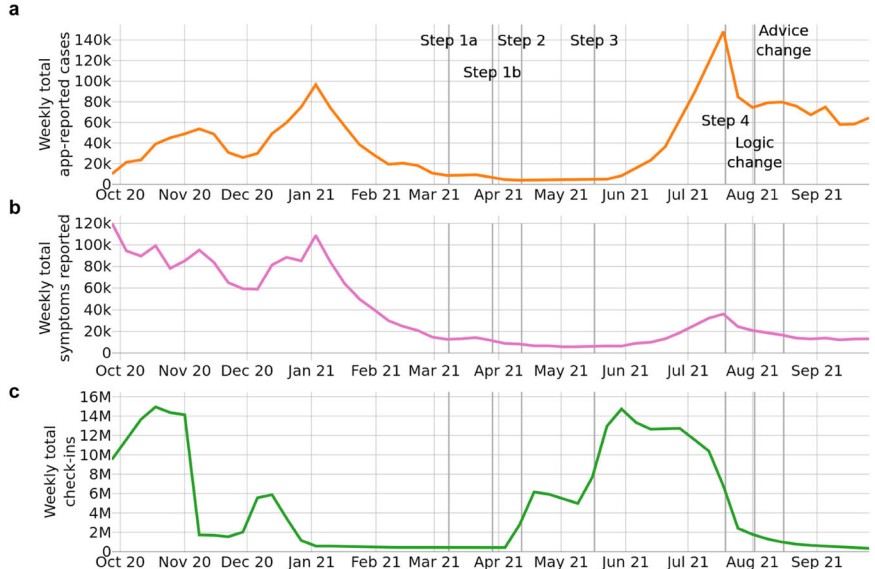

**Fig. 2 | App engagement.** Weekly numbers of **a** app-reported cases, **b** individuals reporting symptoms through the app, and **c** check-ins via the app's QR code functionality. Annotations refer to: the steps of a 'roadmap out of lockdown'[16]; a change to the contact tracing logic of the app for the contacts of asymptomatic cases, and a policy change where some (mainly vaccinated) users were advised to take a PCR test rather than self-isolate upon notification. These events are described in more detail in a timeline in the Supplementary Materials.

detailed description of how user engagement with the app varied over time, and analysis of the relative increase in the probability of testing positive when recently notified. We adapted the modelling approach of Wymant & Ferretti[10] for estimating cases, hospitalisations and deaths averted, building upon the approach to incorporate the background of changing epidemic dynamics including emerging viral variants, population-level restrictions and vaccination roll-out. Our study adds to the body of evidence which shows that digital contact tracing apps have major potential for reducing transmission of SARS-CoV-2 when combined with strong user engagement[11–14].

## Results

### App use and engagement

Following the launch of the NHS COVID-19 app on 24 September 2020, the number of active users (devices with the app installed and an internet connection) increased in a matter of days to over 10 million. Before the release of Version 4.1 on 17 December 2020, app usage data suffered from fluctuations caused by missing or duplicated packets, as can be seen from Fig. 1a. Within a few days of the release of Version 4.1 the recorded number of active users stabilised at around 13.5 million, which is 23% of the total population, or 29% of the eligible (over 16) population using ONS population estimates[15]. Of the active users,

between 71 and 88% had the Bluetooth contact tracing functionality enabled, with this proportion broadly decreasing over the year (Fig. 1a). There was considerable geographic variation in app uptake as seen in Fig. 1b which shows the number of active users as a proportion of the total population for each Lower Tier Local Authority (LTLA) of England and Wales.

Restrictions were gradually eased through the spring and summer of 2021 according to steps of a 'roadmap out of lockdown'[16]. At Step 1b it became mandatory to provide details to NHS Test and Trace when entering some public venues, with QR code check-ins via the NHS COVID-19 app a convenient way to do this; this step was followed by a rapid increase in app check-ins (Fig. 2). This change appeared to drive uptake of the app, with the number of active users reaching 18 million in early July 2021 (38% of the eligible population), while the number of devices with contact tracing enabled peaked at 13.9 million in late June 2021 (29% of the eligible population). There was a consistent decrease in these measures from that point, possibly as a result of the large number of notifications in June–July 2021 which attracted negative media attention and the coining of the term 'pingdemic'. At the end of the period of study on 24 September 2021 the number of active users was 14.6 million (31% of the eligible population) and the number with contact tracing enabled was 10.6 million (22% of the eligible

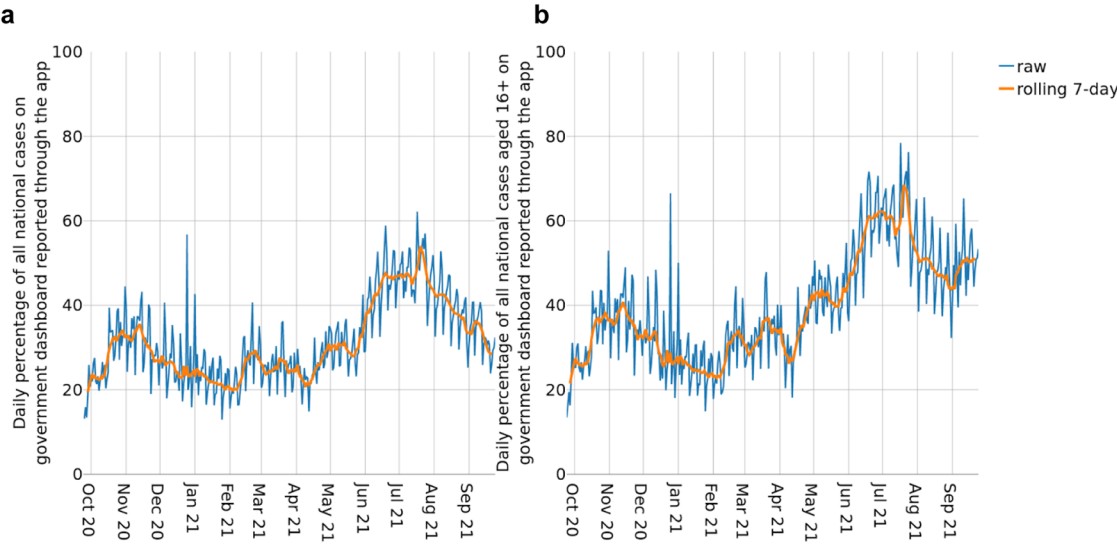

**Fig. 3 | Positive tests reported through the app. a** The percentage of all positive tests by specimen date reported in England and Wales that are app-reported cases. **b** The estimated percentage of positive tests by specimen date amongst those aged 16 or over in England and Wales that are app-reported cases.

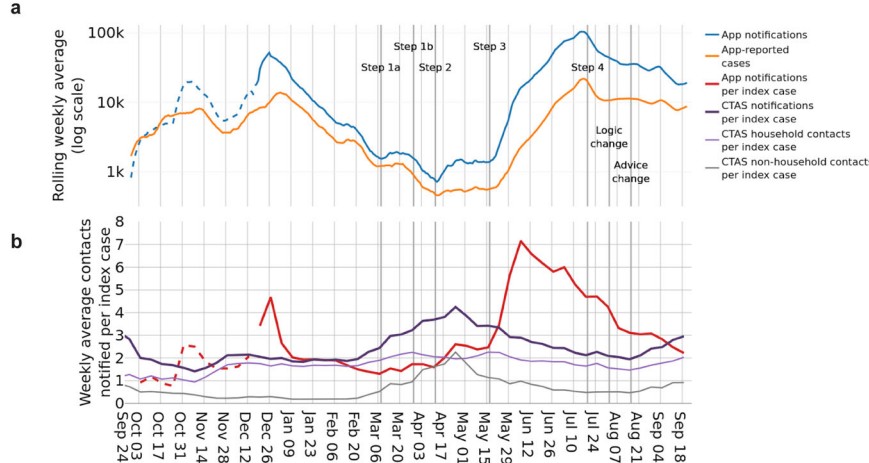

**Fig. 4 | Notifications and positive tests via digital and manual tracing. a** Weekly rolling averages of app-reported cases and the notifications they trigger, shown on a logarithmic scale. **b** Weekly averages of contacts notified per index case via the app and via manual tracing (as recorded by CTAS). CTAS contacts are also shown disaggregated by whether they are household or non-household contacts; the household status of some contacts is not recorded so these values do not always sum to the total number of CTAS notifications per index case. The overlap between contacts traced via the app, CTAS, or both, is unknown. Dotted lines represent estimated values. Annotations refer to: the steps of a 'roadmap out of lockdown'[16]; a change to the contact tracing logic of the app for the contacts of asymptomatic cases, and a policy change where some (mainly vaccinated) users were advised to take a PCR test rather than self-isolate upon notification. These events are described in more detail in a timeline in the Supplementary Materials.

population). Figure 2 illustrates how user engagement with app functionalities varied considerably over the course of the year.

Figure 3 illustrates in more detail how the number of app-reported cases (positive test results received via the app or manually entered into the app) tracked the overall case numbers in England and Wales. Figure 3a shows the number of app-reported cases in England and Wales as a percentage of overall recorded cases. This measure gives an indication of how effective the app can be expected to be in reducing national case rates. Figure 3b shows the same measure but restricted to cases eligible to use the app—those aged 16 or over—which provides a better reflection of public engagement with the app. We find that engagement with test-logging in the app varies over time, and we hypothesise that this engagement depends upon a combination of the number of active users, public perception of the app, and the demographics of cases. For example, engagement will naturally be lower when cases are concentrated amongst school children who are

ineligible to use the app, as demonstrated by the different trends in Fig. 3a, b in September 2021.

## Exposure notifications

Figure 4a shows the number of exposure notifications over time (blue), and demonstrates how this broadly follows the number of app-recorded cases (orange). In total there were approximately 2,138,000 app-reported cases and 7,005,000 notifications during the period of study. The number of exposure notifications per app-recorded case (Fig. 4b, red) averaged 3.28 over the course of the first year. This measure is affected by the number of app users with Bluetooth contact tracing enabled, by contact rates amongst app users, by the risk scoring algorithm[17] and risk threshold of the app, and by the proportion of test-positive app users who consent to contact tracing after recording their positive result. The risk threshold of the app was lowered in app Version 3.9 on 29 October 2020 and again in Version 4.1

on 17 December 2020 to increase the number of contacts to be notified. (Note that not all devices upgrade to the latest app version as soon as it is available.). The proportion of test-positive app users consenting to contact tracing has varied over time but we estimate it to be between 40 and 55% (Supplementary Fig. S1).

## Digital and manual contact tracing

Figure 4b demonstrates the scalability of digital contact tracing. The app and manual contact tracing (documented by CTAS[18]) work in complementary ways; the number of contacts identified per case (Fig. 4b, purple) for each method varies according to a number of factors. One class of factors is social restrictions (enforced or voluntary), for example manual contact tracing is expected to reach more contacts when there is a larger proportion of contacts that are within households, schools and nurseries, whereas the app is expected to reach more contacts when there are many interactions between individuals aged 16 or over outside the home. In particular, the app can reach contacts for whom the individual testing positive does not have contact details and/or does not recall meeting, such as on public transport[14]. Another class of factors is overall case numbers and contact rates. The app is able to process exponentially increasing numbers of positive tests and/or high contact rates, whereas manual tracing has a reach determined by staffing numbers that varies over time. In fact, manual tracing shows an anticyclic trend in Fig. 4b, with reduced numbers of contacts per case reached during periods of high case load. This trend occurred despite the gradual introduction of automated tools to support manual contact tracing, with cases being automatically sent a form to complete, and all the contacts for whom they provided a phone number sent automatic text messages; this system was fully implemented by 17 March 2022 (after the period of study) in Wales[19]. Routine contact tracing ended in England on 24 February 2022; in the final week, 68% of recent close contacts were reached[18].

## Testing positive after exposure notification

To be useful in contact tracing, the app needs to send notifications to individuals who may be infected. At the point of notification, the infection status of notified individuals is not recorded in app data. However, when a user enters a positive test result, the data indicates whether or not they were asked to isolate shortly beforehand (Fig. 5a, b). This allows estimation of the proportion of individuals who report a positive test after receiving an exposure notification (TPAEN) during the recommended isolation period or in the 14 days after its end. The proportion TPAEN represents a lower bound on the proportion of notified individuals who are infected, since not all infected individuals will apply for a test and report their positive result through the app. The Office for National Statistics estimated that the number of individuals infected was 1.5 to 3 times higher than the number of cases reported nationally during the period of study[20]. We previously estimated the proportion TPAEN to be 5–7% during the period October to December 2020[10]; our estimate of how it subsequently varied over time is shown in Fig. 5c. Variation over time is likely due to changing vaccination levels, viral variants, policy and behaviour (e.g., indoor versus outdoor contacts). It must be emphasised that, unlike in many other countries that deployed exposure notification and other forms of contact tracing, policy in England and Wales did not always include a recommendation for notified individuals to take a test, and so the proportion TPAEN is not comparable to other countries. Asymptomatic but contact-traced individuals were ineligible for a free PCR test until Spring 2021. App notifications changed to recommend booking a test following a venue alert and following an exposure notification in Versions 4.6 (10 March 2021) and 4.9 (27 April 2021) respectively. From 16 August 2021, app Version 4.16 reflected national policy that some contact traced individuals were no longer asked to isolate but were recommended to take a test. More details are provided in the Timeline

in the Supplementary Materials. We note that each of these changes may have driven more case-finding in app-notified individuals.

In Fig. 5d–f we compare case numbers in recently notified app users to case numbers in two observational control groups for which data was available to us. Together these measures provide some indication of the accuracy of app notifications in alerting users at high risk of having been recently infected. First, Fig. 5d shows that notified app users were more likely to report a positive test than a randomly selected person over 16 from England was to test positive, by a factor of at least 2 at all times throughout the period of study. Lower values from June 2021 (Fig. 5d) may be attributed to higher SARS-CoV-2 prevalence and lower user engagement with the app (Figs. 1–3). The highest value was in May 2021 when notified app users were 26 times (16–46) more likely to test positive than the general population. Second, Fig. 5e, f compares case numbers in recently notified and not-recently-notified app users. Figure 5e shows that the proportion of recently notified app users testing positive (green) is consistently higher than the proportion of not-recently-notified app users testing positive. The ratio between these groups of the odds for testing positive is shown in Fig. 5f: it was usually above 3 (lowest point 2.4 (2.3 to 2.6)), and at the peak in May 2021 it was 77 (54, 106). Figure 5e shows that the reduction in accuracy can be attributed mostly to the increase in the number of cases in the population, rather than to changes in app sensitivity. The decreases observed in June–July 2021–around the time of high numbers of notifications commonly referred to as the "pingdemic"–could also be at least partly attributed to users deleting the app after exposure notification and before testing positive (a phenomenon which could affect all time periods but is likely to increase with negative media attention), but further data would be needed to confirm this.

## Cases, hospitalisations and deaths averted

Finally, we estimate the number of cases, hospitalisations and deaths averted by the app over the course of its first year (Fig. 6). Our confidence intervals are large because we rely on an estimated TPAEN, estimated delays, estimated overlap with other means of discovering infection, and estimated levels of adherence to app guidance. We estimate that the app's contact tracing function averted 1 million cases (sensitivity analysis 450,000–1,400,000), corresponding to 44,000 hospital cases (SA 20,000–60,000) and 9600 deaths (SA 4,600–13,000) over the course of its first year. Figure 6d, e demonstrate the variation in the app's epidemiological impact across LTLAs of England and Wales. This heterogeneity reflects not only uptake (cf Fig. 1b) but also engagement with reporting positive tests through the app and consenting to contact tracing, clustering of app users, and prevalence amongst app users at times of high viral prevalence.

## Discussion

In this study, we presented data concerning public engagement with the NHS COVID-19 app in England and Wales throughout its first year, and the epidemiological impacts of its digital contact tracing functionality. The app can be evaluated based on the five key epidemiological and public-health requirements for SARS-CoV-2 contact-tracing apps identified by Colizza et al.[8]. First, integration with local health policy: the NHS COVID-19 app was developed by NHS Test and Trace, now part of the UK Health Security Agency, and is aligned with local health policy. That is, the app is frequently updated so that public health advice and legal requirements are matched by both the general information provided to all users as well as the specific advice given to notified users. These varied considerably during the Autumn-Winter 2020 "Tiers" system, and have often differed between England and Wales. Second, high user uptake and adherence: uptake of the NHS COVID-19 app, though geographically varied, represents a high proportion of the population compared to other SARS-CoV-2 contact tracing apps in Europe[21]. Although we know daily totals of app

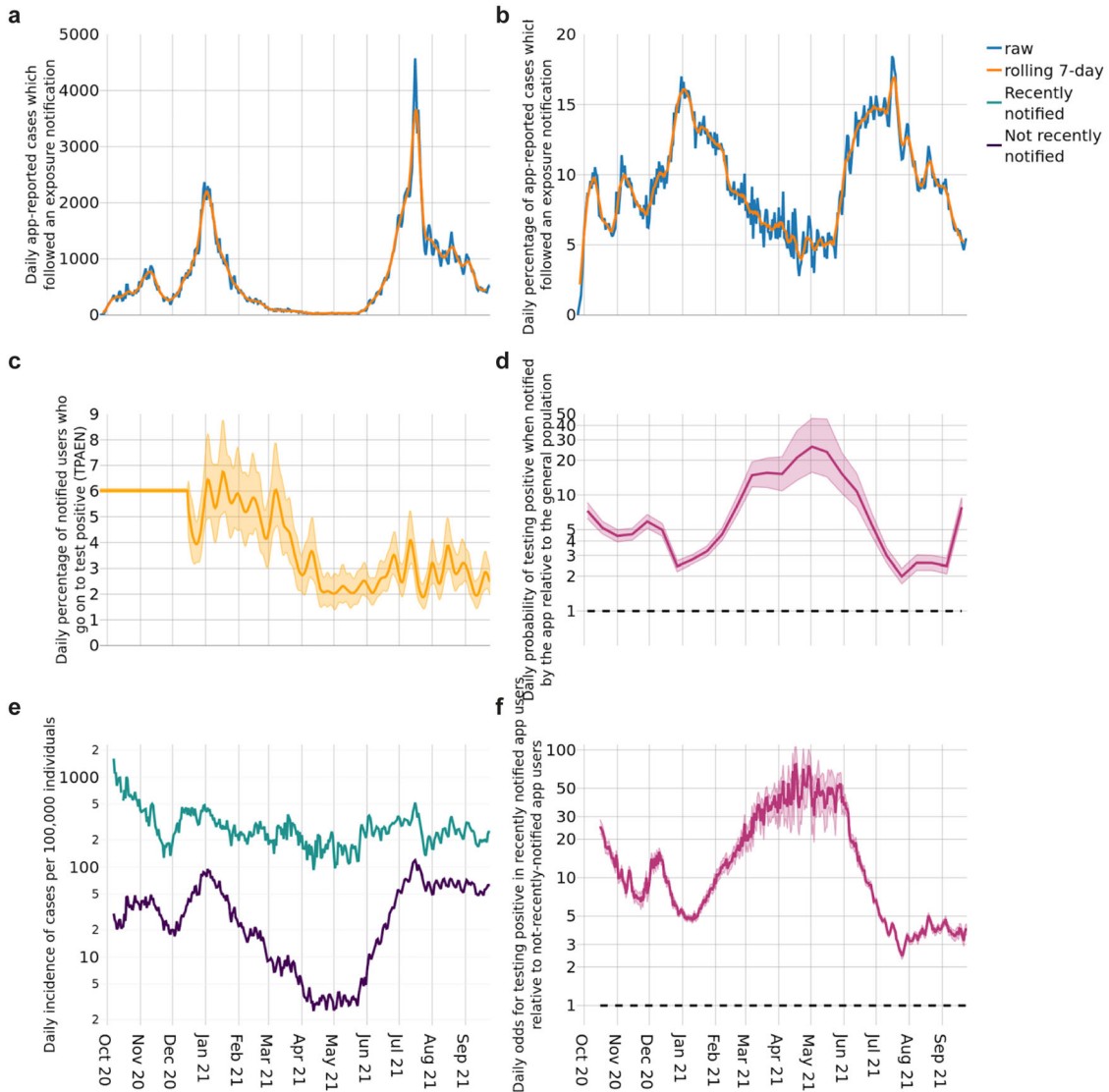

**Fig. 5 | Testing positive after exposure notification. a** The daily number of app-reported cases which followed an exposure notification, and rolling 7-day average. **b** The daily and rolling 7-day average percentage of app-reported cases which followed an exposure notification. **c** The mean estimated proportion of notified individuals who enter a positive test result into the app shortly after exposure notification (TPAEN). Shading around the line indicates the 95% credible interval. **d** The probability of testing positive after notification relative to a random member of the 16+ population, shown on a logarithmic scale. The central estimate and shading around the line correspond to the uncertainty reported for the ONS infection survey (official estimate and 95% credible interval respectively). The horizontal dotted line indicates a value of 1, i.e., equal probability. **e** Daily new app-reported cases per 100,000 active users by notification status. **f** Daily maximum likelihood estimates of the odds for testing positive in recently notified app users relative to not-recently-notified app users. Shading indicates 95% confidence intervals. The horizontal dotted line indicates a value of 1, i.e., equal probability.

notifications, data concerning adherence to app advice is limited; the ONS estimated adherence to be relatively high based on small sample sizes, and we assume more conservative estimates. Third, quarantining infectious people as accurately as possible: we present the first detailed longitudinal analysis of the NHS COVID-19 app's accuracy, providing comparisons with the general population and between recently notified and not-recently-notified app users. Fourth, rapid notification: notifications are usually within 4 h, as long as the device is switched on. They rely on quick test turnaround times for greatest success; typical test result waiting times are provided by UKHSA[18], shown in Supplementary Fig. S3. Fifth, the ability to evaluate effectiveness transparently: we estimated the number of cases, hospitalisations and deaths averted by the app with the aims of informing app development and supporting transparency.

We assessed the epidemiological impact of the core functionality of the NHS COVID-19 app: digital contact tracing. This functionality

acted in conjunction with other services available through the app (accessing local area information, venue check-in, symptom checker, test ordering, self-isolation countdown following a positive test, and access to self-isolation payments) but we do not evaluate those other services here because of limited insight from app analytics data. The increase in app uptake after the requirement for venue check-in was introduced indicates the potential importance of integrating different digital tools for maximum user engagement and effectiveness. A broader international review of the use of digital tools in the SARS-CoV-2 pandemic is given by Pandit et al.[22].

Our analysis of the app data was limited by its anonymised, aggregated and minimal nature. We use a conservative approach wherever possible, noting that TPAEN is likely to be an underestimate of the proportion notified who are infected, and we are likely underestimating the case hospitalisation rate for the Alpha wave. Our estimates of the timings between an app notification and discovering

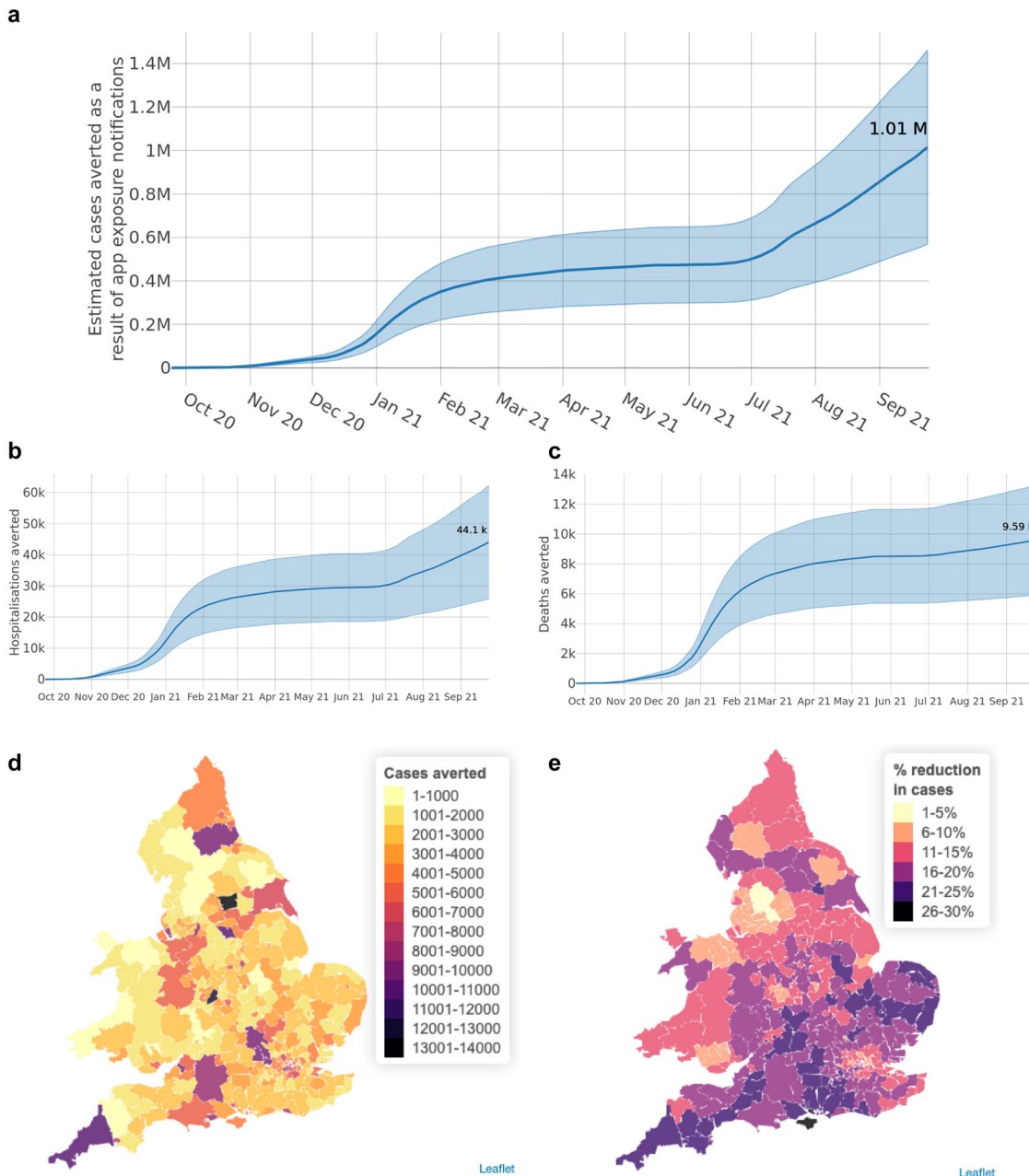

**Fig. 6 | Epidemiological impacts.** Cumulative estimated numbers of **a** cases, **b** hospitalisations and **c** deaths averted by app exposure notifications between 24 September 2020 and 24 September 2021. Shading in panels **a**–**c** indicates the range of outcomes between upper and lower plausible estimates of an individual's reduction in risky contacts as a result of receiving an app notification, while the central estimates correspond to moderate reductions in risky contacts. **d** Estimated cases averted in each LTLA. **e** Estimated percent reduction in cases in each LTLA.

infection via another means are poorly informed by data, as are our estimates of adherence levels to app notifications, and so we considered a wide range of plausible values for these parameters. We include a sensitivity analysis in the Supplementary Materials. In our analysis of hospitalisations and deaths averted we assume that app users together with their onward chain of contacts are representative of the population of England. This assumption is unlikely to be true for short transmission chains, for which individual characteristics are expected to be close to those of the app-using population rather than the general population. It becomes better justified when the initial case directly averted by the app occurs nearer the start of a wave—allowing for more population mixing as the counterfactual transmission chain proceeds through more generations—and these larger chains contribute more to the total result. We implicitly assume that the entire

onward transmission chain remains in the same LTLA as the notified app user. This is unlikely to be true in general and is a limitation of our approach. However, we note that this assumption is better justified during the "Tiered" social restrictions of Autumn–Winter 2020, when there was more heterogeneity between case numbers across LTLAs, whereas later in the period of study when there was more freedom of movement there was also more homogeneity of case numbers across LTLAs. Finally, we also assume that the individual risk of getting infected during each wave was small, neglecting risk saturation for repeatedly exposed individuals.

When we calculate the numbers of cases, hospitalisations and deaths averted we are implicitly comparing to a counterfactual scenario where the app is not present but all other interventions and behaviours remain unchanged. The *potential* impact of the app is

higher than these values because we only attribute credit to the app in changing behaviours in the period after receiving an app notification and before learning via other means that one is at high risk of being infected. For example, it is plausible that an individual who develops symptoms will take more care to restrict their contacts if they also had an app notification than if they had symptoms alone, but here we use the conservative approach of not attributing to the app any reduction in infections after the onset of other 'alert' mechanisms.

Further, the estimates we present for the total cases, hospitalisations and deaths averted are likely to be underestimates because we do not incorporate indirect effects of app usage beyond contact tracing, as should have been captured by the spatial statistical analysis in Wymant & Ferretti[10]. In that instance, the spatial analysis estimate was slightly more than double the modelling estimate, which could also have been due to residual confounding as well as indirect effects. We found it was not possible to extend this approach to the whole of the first year because of higher geographical mixing as restrictions were lifted in spring 2021 (the app does not track user locations), and the confounding effects of immunity via prior infection and vaccination. Possible indirect effects of the app which are not captured in our approach include: pre-emptive behavioural effects, where app users reduce social interactions to lower the risk of notification; the network effects of app usage, which mean that as well as being less likely to pass on an infection, app users are in fact less likely to become infected because they may be warned to self-isolate when the virus is circulating in their close contact network (more details in the Supplementary Materials); and the non-contact-tracing functionality of the app, including ease of accessing information and being able to quickly book a test. In particular, it would be interesting to assess the full impact of the large numbers of notifications (or "pings") in June and July 2021, referred to by the media as the "pingdemic". This was immediately followed by a steep decline in the number of cases, and a causal link has been hypothesised[23]. We also have not considered any potential wider indirect impacts of the app on policy, implicitly assuming that, without the app, all other interventions would have been identical: we do not, for example, model a counterfactual scenario where a lockdown may have been implemented differently. These limitations present challenges for using these results in onward analyses such as cost-benefit analyses.

Notwithstanding these limitations, we conclude that digital contact tracing has played an important role in reducing transmission of SARS-CoV-2 in England and Wales in practice, as was expected in theory. The NHS COVID-19 app experienced high user engagement, identified infectious contacts well, and helped to avert appreciable numbers of cases, hospitalisations and deaths. The effect of digital contact tracing apps can be improved by increasing uptake, increasing app-recorded cases consenting to contact tracing, and increasing adherence to advice to self-isolate and/or take a test[24]. We conclude that digital contact tracing—a relatively low cost and rapidly available intervention—is a valuable public health measure for reducing transmission in any future epidemic waves of SARS-CoV-2 or other applicable pathogens.

## Methods

We use publicly available data and anonymised, aggregated app data, described in brief below and in detail in the Supplementary Materials. The Google Apple Exposure Notification protocol permits collection of some anonymous data packets, and the design of the app, including its data collection, were approved by Google, Apple and the UK Information Commissioner's Office.

### Data on numbers of app users and uptake

Figure 1a presents data from UKHSA[25]. The number of "active users" is given by the daily number of devices with the app installed that sent a data packet to the central servers, indicating that the device was switched on and had internet access at some stage during the day. This includes users who paused the Bluetooth contact tracing functionality of the app at some stage during the day. The number of users with contact tracing enabled is the estimated number of users with the app installed and where the app is deemed 'usable' (app version supported and onboarding completed) and 'contact-traceable' (Bluetooth enabled and able to receive notifications). In Fig. 1b we use daily app analytics data packets aggregated to the LTLA level. To calculate uptake as a proportion of the population we use ONS population estimates by LTLA[15]. Information on postcode area (and, in some cases, LTLA) was entered by users; the app does not track users' location.

### Public data on app use and total cases

For Fig. 2 we use data from UKHSA[25] for the numbers of app-reported cases, symptoms reported through the app, and check-ins via the app's QR code functionality. We aggregate to the total England and Wales level. We also use publicly available data on the number of cases across England and Wales[26]. Note that public app data is subject to small number suppression[27].

### Data on notifications and app-reported cases via the app and via manual tracing

We use the daily number of devices which received an exposure notification as a measure of newly notified users. An individual device cannot receive multiple exposure notifications within a short time: after an exposure notification is received, no further exposure notifications can be received until the end of the isolation period plus a 14-day window. The release of app version 4.1 on 17 December 2020 provided further app data fields and improved the accuracy of exposure notification data—we provide full details in the Supplementary Materials. There is also daily data on the number of app-reported cases. App-reported cases include positive results of tests (PCR and LFD) ordered through the app, which are automatically recorded, as well as tests accessed in other ways (for which the individual is sent a code with their positive result, and is encouraged to enter it into the app). When a user records multiple positive results within a day, this is only counted once in our data. However, for privacy reasons, it is not possible to distinguish if a user enters positive tests over multiple days, and may therefore be counted repeatedly. This may somewhat distort our comparison to national daily case data which is deduplicated; any such effect is likely to increase over the period of study as LFDs became more widely available. For manual (non-app) contact tracing, we use data from the Contact Tracing and Advice Service (CTAS)—the web-based tool to record information about people in England who have tested positive for SARS-CoV-2 and their contacts[18].

### Estimating the proportion testing positive after exposure notification

If an individual tests positive and reports this through the app, it is possible to see from the data whether that individual had already received an exposure notification in the recent past, that is, whether the positive result was reported during the recommended isolation window or within the 14 days afterwards. We use this data to determine the proportion of app-recorded cases who were pre-notified, and from this to estimate the proportion of notified individuals who go on to test positive. It is likely to be an underestimate of the true proportion of individuals infected (i.e., the secondary attack rate), because not all users will apply for a test and report their positive result through the app, and some will report a positive test before being notified.

The proportion TPAEN will reflect the type of contacts people have (e.g., it may be higher for indoor contacts than outdoor), the infectiousness of the index case (which can vary according to the variant, and possibly by demographics such as age of index case) and the susceptibility of the index case's contacts (vaccinated individuals will be less susceptible than those not vaccinated, particularly in the

pre-Omicron time period of this study). We do not report the proportion of test results that are positive, since negative test results were reported through the app at very different rates in different time periods.

Data limitations before 17 December 2020 dictate that, for this time period, we rely on a fixed estimate of the proportion testing positive after exposure notification (TPAEN) from Wymant & Ferretti[10]. After this date we use a Bayesian estimation that accounts for delays between notification and test positivity. Additional app data, separate from the daily analytics packets, provides the empirical distribution of the time taken from notification to reporting a positive test, when both of these events occur. Briefly, we model the expected number of actual notifications and the proportion TPAEN as exponentials of natural splines with weekly knots in order to provide a smooth estimate. Spline coefficients are products of relative coefficients and absolute coefficients. The expected daily number of positive tests after notification is computed by multiplying the expected number of notifications by the proportion TPAEN and convoluting this time series with the delay distribution. The time-varying proportion TPAEN is then estimated by comparing observed and expected daily counts, modelling the former as negative-binomially distributed about the latter. Further details are provided in the Supplementary Materials.

We provide context for the proportion TPAEN by comparing it with the proportion of all individuals across England and Wales testing positive, i.e., with prevalence. For an app user who is notified, we estimate how much more likely they are to report a positive test than a person randomly selected from the population of England aged 16 or over to test positive, using ONS data[15, 20], see Supplementary Materials for more details. ONS values refer to infections occurring in private households and exclude infections reported in hospitals, care homes and/or other communal establishments. Comparable data is not currently available for Wales but we note that 96% of app users report having an English postcode district.

We analyse the relative incidence of app-reported cases amongst recently notified and not-recently-notified app users, and estimate the odds of reporting a positive test in the app for the former group relative to the latter group using a separate logistic regression for each day, with recent-notification-or-not as the only predictor variable.

### Estimating the number of cases, hospitalisations and deaths averted

To estimate the number of cases averted we use a method adapted from the modelling approach of Wymant & Ferretti[10]. We model the number of cases averted as the daily product of five factors: (i) the number of notifications, (ii) the proportion of those notified who are infected, (iii) the fraction of the infectious period which occurs between an individual receiving an app notification and before discovering their infected status via another means, (iv) the individual's fractional reduction in "risky contacts" as a result of an app notification, and (v) the expected size of the onward transmission chain which would have originated from the contact had they not been notified.

We use TPAEN as an estimate of factor (ii), whilst noting that it is likely to be an underestimate. For factor (iii) we note that the delay between being exposed to the virus and receiving an exposure notification varies between individuals and over calendar time. We use app analytics data to estimate the average daily delay between exposure and notification. The overlap and relative timing of app notifications compared with an individual suspecting or discovering their infectious status by another means (word of mouth between friends or household members, manual contact tracing, symptoms, testing positive, etc.), and the extent to which these caused a change in behaviour, are poorly understood and we

chose plausible, time-varying values as detailed in Supplementary Table S1. We also provide a sensitivity analysis in the Supplementary Materials to consider a range of assumptions about the interaction of app notifications with other interventions across the epidemic waves. We vary the modelled proportion of app-notified individuals who also discover their probable infected status by another means and the extent to which they reduce their risky contacts at that point. We do not attribute to the app any reduction in risky contacts (and, therefore, any reduction in infections) which occurs after an app-notified individual has further reason to believe they may be infected.

We consider a "risky contact" to be a physical interaction between two individuals where, if one is infectious, the other could become infected if they are susceptible. The "riskiness" of a contact cannot currently be measured with complete accuracy, but is determined by factors which include the proximity and duration of the contact, the ventilation of the space, and the use of personal protective equipment. Supplementary Table S1 also describes the limited data available to inform this risky contact reduction (factor iv above). We do not assume that all notified app users will perfectly follow app guidance concerning self-isolation and testing. Further, we assume that many app users will have regular and unavoidable risky contacts within their household setting, which are unchanged by an app notification. Following Wymant & Ferretti[10] we assume that the average user reduced their risky contact levels by 60% (lower 38%, upper 82%) after a notification in the period before the Delta variant (24 September 2020 to 17 May 2021). We assume a 40% reduction (20%, 60%) after that to account for changing public opinion and policy, particularly the policy change of 16 August 2021 which advised users who were fully vaccinated or aged 16–18 to seek a test rather than self-isolate. These central estimates and upper and lower bounds are illustrated by the lines and shaded regions respectively in Fig. 6a–c. Finally, for (v) we split the data into three "waves": a pre-Alpha variant wave, an Alpha variant wave, and a Delta variant wave. The counterfactual "onward transmission chain" is modelled as a proportion of the cases remaining in the current wave; transmission chains are considered to start afresh with each wave. Further details are provided in the Supplementary Materials and Fig. S2.

To estimate the numbers of hospitalisations and deaths averted by app notifications, we multiply numbers of cases averted by the observed fractions of cases that were hospitalised or died, respectively. We use the empirical case hospitalisation rates of 9.8% for the pre-Alpha wave, 5% for the Alpha wave and 2.7% for the Delta wave estimated from government dashboard data[26]. The estimate of 5% for the Alpha wave is likely to be an underestimate as it was measured from April 2021 data, when vaccinations were having the greatest impact in the wave. We use estimated case fatality rates of 1.5% for the pre-Alpha wave, 1.9% for the Alpha wave and 0.2% for the Delta wave[28]. These hospitalisation and fatality rates were based on the full population of England. By applying them to averted cases among app users, we are assuming that the subpopulation of app users and their onward chain of contacts (including contacts of their contacts etc.) is similar to England as a whole with regard to susceptibility to severe clinical outcomes given infection.

### Reporting summary

Further information on research design is available in the Nature Portfolio Reporting Summary linked to this article.

## Data availability

Data access is managed by UKHSA, who will make available on request the data needed to replicate this analysis, via the UK Data Service. Access is controlled for privacy reasons.

## Code availability

Code to reproduce these results, which can also be run on the dummy data provided, is available from https://github.com/MichelleKendall/epi_impacts_NHS_COVID-19_app_first_year[29].

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

## Acknowledgements

We are grateful for the help and support from teams across UKHSA, Zühlke and previously at NHS Test and Trace. In particular we thank the NHS COVID-19 app Data and Analytics Team for their invaluable support with data access, management and analytics. This work was funded by a Li Ka Shing Foundation award and research grant funding from the UK Department of Health and Social Care (DHSC), both to C.F., and by the National Institute for Health and Care Research to the Health Protection Research Unit in Genomics and Enabling Data, grant number NIHR200892, for M.K. and X.D. The views expressed in this article are those of the author(s) and are not necessarily those of UK Health Security Agency (UKHSA) or the Department of Health and Social Care (DHSC).

## Author contributions

All authors (M.K., D.T., C.W., A.D.F., Y.B., X.D., L.F., and C.F.) contributed to the conceptualising, writing, and reviewing of this manuscript. M.K., D.T., C.W., L.F., and C.F. did the analyses.

## Competing interests

M.K. has a data sharing agreement with UKHSA. DT was an employee of Zühlke which provided consultancy to UKHSA. C.W., L.F., and C.F. were named researchers on a grant from DHSC to Oxford University. A.D.F. and Y.B. were employees of UKHSA. X.D. declares no competing interests.
