## [Peer Review File · Nature Communications]

Epidemiological impacts of the NHS COVID-19 app in England and Wales throughout its first yearREVIEWER COMMENTS

Reviewer #1 (Remarks to the Author):

The paper uses publicly available data from the NHS about the use of its exposure notification app to report on the epidemiological impact of the app. More specifically, it tracked the app reported cases, whether notified users ended up submitting a positive test and finally it gave an estimate of the number of averted cases, hospitalizations and deaths.

I believe that the paper provides an important data point on the impact of digital contact tracing. However, I have struggled with identifying its novelty. I think the longitudinal aspect of the study is indeed interesting, but the findings are not novel. The fact that digital contact tracing can be/was effective has been established by a number of studies, including a couple by the authors. Hence, this paper is not necessarily adding significantly to the existing art.

The period with excessive notifications is quite interesting as such bugs can negatively impact the public perception of the app. Hence, it would have been really interesting to look into how that bug affected the adoption and use of the app.

The analysis of TPEAN is one of the most novel parts of the paper. Unfortunately it wasn't developed further. TPEAN varied over time. The authors rightly attributed that to changing vaccination levels, viral variants, policy, and behavior. Nevertheless, the paper did not attempt to find causal links between these factors, and the observed variability.

While I acknowledge the difficulty and limitations imposed by the aggregated and anonymized nature of the dataset, I believe that augmenting it with other data sources like surveys and data on manual contact tracing or even anonymized app data from other countries could have helped improve the value of this work.

Reviewer #2 (Remarks to the Author):

In this paper the authors present an assessment of the epidemiological impact of the NHS COVID19 app in England and Wales, one year after its deployment, and can be seen as a follow up work after [Wymant et al, Nature 2021] that accounted for initial impact of the app 3 months after deployment. This follow-up paper is based on the former to quantify the impact of the app but adds a range of new insights, including (i) impact of a modification in the app settings, (ii) longitudinal analysis and effect of different variants, variation of uptake based on the app being needed to produce necessary QRs, user fatigue, vaccine roll out, etc.

The main results depicted in the paper include (I) observing a clear signal of the app sensibility (users who were recently notified and get tested are more likely to get a positive test than those who were not recently notified) and (II) an assessment of the impact of the app in terms of transmission averted (about 1 million cases, translating into circa 44k averted hospitalisations and circa 10k deaths). This, together with the fact that app deployment is a rather cheap public health intervention, confirms its usefulness for pandemic response.

The paper is clearly written, the methodology is well documented and the results very impactful. While this is indeed a follow up work that confirms early results by the same group, I believe it is still a much needed piece of work. The reason is because the data and analysis coming up from the NHS app deal with a scenario where the app uptake is relatively high --this being a clear bottleneck both in theory and in practice, as documented by both modelling and experiments in recent literature--. It seems the NHS case can provide evidence of how a digital contact tracing technology could effectively contribute if (and this is a big if) adoption (user uptake) is sufficiently high and stable over time.

Accordingly, in general I recommend publication of this work in Nature Communications. Below I summarise a list of aspects which I believe could benefit from a more detailed explanation or consideration. The comments are not sorted in order of importance.

- The national uptake is given in terms of millions, might be a good idea to also give it in terms of percentage of the whole population so that results can be used for extrapolation and comparison with other works?

- Comments on Fig1:

Panel A of Fig 1 displays the (geographically-aggregated) number of "active apps" throughout time, whereas panel B of the same figure depicts a geographically disaggregated yet time-aggregated visualization. I have several comments and questions on this Figure.

* How does the data collection and its granularity allow you to determine that the app is actually "active" wrt, say, a situation where the user downloaded the app but then uninstalls it or switches off the exposure notification system? I know that some explanation is reported in the Methods Section, but I think it would be helpful to clarify how this data was collected and to which extent this was compatible with the rather strict privacy-preserving paradigm. Note this type of data was not available "by design" in other contact tracing apps.

* At a first order approximation, one could argue that population density might be roughly correlated with the R number (e.g. more populated regions might tend to yield more densely connected social contact networks, or might be prone to host super spreading events, etc). If true, this assumption would imply that from a transmission viewpoint it would be more critical to have a larger app adoption/adherence in those more densely populated regions. I wonder if the authors have considered or taken into account this aspect in their estimation analysis. Spin-off analysis could also include checking whether there is a correlation between more densely populated regions and a higher adoption (visually it seems to be the case from the panel b?), correlation between app adoption (% population) and ICU density, and many others.

- Comments on Fig 2 and 3:

Results seem a priori impressive, where the percentage of positive tests associated to app users that had been notified increased up to 70%. This is, however, just a side of the story, and I am not sure the authors have presented this story in the most balanced and clear way. I'd suggest, if possible, to estimate some proxy of a confusion matrix, that would provide not only true positive rates but also false positive rates, etc. It is expected that the false positive rate is large (somehow related to the pingdemic concept), and the authors should discuss this aspect (for instance, they mention the number of cases detected increased after a change in the app settings, does this translate in a stable or increased false positive rate?) I think it is only fair to provide all these aspects of the data.

- Abstract and Figs 5

One of the main results is a positive association between being notified by the app and testing positive, as compared to those not notified by the app. It is not clear to me what the control group is: is it a group of people that decided to get tested (because of whatever reasons but excluding app notification)? Or a random selection of people that were tested (like a random sampling)? I think this is important to clarify because the conclusions are different in both cases. From the abstract I had the impression it's the former, but from the text I then thought it's the latter. If it's the latter, the results are nice but somehow expected. If it's the former, the results are much stronger because of the following (I'm slightly reaching, just for the sake of argument): one could argue that those citizens that adopt the app might be more sensitised to the overall pandemic situation, and that in turn might include (i) take lesser risks, (ii) "follow the rules" more strictly, and so on. Conversely, those users not engaged with the app might tend to be more relaxed. In that scenario, those citizens not using the app and yet voluntarily deciding to get tested would be more likely to get a positive test, if only because these more relaxed citizens might only go through the burden of getting tested when they have a clear case (having clear symptoms, having a positive case in the household, etc). Under this reasoning the result of the authors would be even stronger.

Is it possible that the authors address both scenarios in panels of Fig 5?

A clarification would be needed.

- Transmission mechanism and rationale for DCT.

In the early times of Covid digital contact tracing, when it was tacitly assumed that droplets were the main transmission mechanism, it seemed sensible to impose a 1.5 metre distance + some time window (15 min) as the threshold to consider a potential contact was established. Actually, these parameters were really unknown, and the truth was that the 1.5 metre part was more a criterion

based on bluetooth quality (beyond 2m the signal intensity decay is not reliably correlated with actual distance). Even worse, with the realization that the virus has an airborne transmission, the 1.5 metre criterion seemed more arbitrary (or just based on technological considerations). In a nutshell, the whole criterion to determine when a contact has been established is, I believe, a very complicated issue with not obvious response from an infectious disease point of view. I think this aspect should be clearly acknowledged and discussed in the paper, as throughout this work (and surely, the rest of the work in DCT) assumes that these criteria are biologically-informed, which is far from clear to be the case. If new evidence has come to light and what I am saying is not true, I would be delighted, either way, I think this merits a discussion, because it is a clear limitation (or a premise of the usefulness of) the whole DCT paradigm.

In this sense, the main part of the paper mentions a change in the detection criteria of the app but (not sure if I missed this) no detailed information of what the change is, and why it is sensible, is provided.

Signed: Lucas Lacasa

Reviewer #3 (Remarks to the Author):

Summary

This paper reports on the NHS COVID-19 app, whose primary function was to notify contacts of recently diagnosed individuals with SARS-CoV-2 infection among app users and advise them to take the action recommended/required at the time (ie, quarantine at home, later, take a PCR test). The paper first describes the app's uptake and engagement, average numbers of notifications, and positive tests among notified users over time reflecting epidemiological dynamics and changing policies on social mixing. The authors then provide an estimation of the number of cases, hospitalisations and deaths likely averted through the app's contact tracing action, based on a number of assumptions, and conduct further sensitivity analyses to vary some of these assumptions.

Overall comments

Thank you for the opportunity to review this paper, which I found very interesting and informative. It is of high interest to the readership of this journal. Descriptions of the performance of COVID-19 tracing apps are important to conduct and report and this paper is thorough and thoughtful in doing so. I have a set of concerns about methods for attributing averted infections due to the app, some comments about report structure and about additional considerations. Apart from a few points about what detail is included where (below), I found the paper well-written and the figures were clear and nicely presented.

Major comments

1. Household versus non-household contacts

My main concern about the estimation of cases, hospitalisations and deaths averted due to the app, is around how household versus non-household contacts are treated. The transmission probability to household contacts is relatively high compared to other contact types, and it seems likely that two app users within a household are likely to be notified, given the probability of close, long-duration and repeated proximity. Household versus non-household contacts are not mentioned much in the paper, but I think there are some important differences between household and non-household contacts that need to be accounted for in assessing the impact of contact tracing. Household contacts are different in that they 1) are not protected from infection by isolation of the infected individual at home (or not as much as non-household contacts) and because 2) tracing household contacts, whether manually or with the app, could be considered to be redundant, for many or most households. The second issue can be dealt with by varying the proportion of case notifications due to the app (as varied in the sensitivity analysis), but I am not sure how the first issue is dealt with in this analysis. The proportion of contacts made during the infectious period who are household contacts, and the proportion of transmission among household versus non-household contacts would also be expected to change over the course of the epidemic

as policies around social mixing changed. Additionally, contacts of within-household cases were for most of this time period meant to begin isolation immediately upon the household index case's symptoms, so if following policy and depending on test delays, they would not have had out-of-household contacts during their infectious period, regardless of whether they are traced.

Presumably, there is not a definite way of identifying when notifications from the app are between household members - could this be clarified? I wasn't quite clear from the supplementary whether data was retained on the duration/repetition of contact which might help to indicate household versus non-household contacts, or, if collected, whether this data was made available for analysis.

I do think the paper needs an explanation for how household versus non-household contacts are dealt with in assessing the impact of the app on cases, hospitalisations and deaths, and perhaps some reconsideration of the results reported as the primary analysis as non-protection of household contacts by isolation of infected individuals (whether identified as such or not) should reduce the proportion of cases that the app could prevent.

I do agree with the point that the app should have been in a better position to identify potentially infectious out-of-household anonymous contacts relative to manual tracing.

2. Level of detail in the main paper versus supplementary

I would have preferred more methodological detail in the main paper itself, not only in the supplement. Information that I would consider highly relevant to the extent to which I trusted findings was in the supplementary section but not in the Methods. In some cases, because the Results is prior to the methods section, I think it would be helpful to include brief sub-clauses defining key terms in the Results when these terms are first used. Specific instances include:

*Results - could a very brief sub-clause defining 'active user' be included in the Results text itself (first paragraph)?

*Results - ditto 'app-reported cases'

*Methods- further clarity about the time-varying parameters underlying the estimation of averted cases, hospitalisations and deaths (eg reflecting different test result return delays), would be helpful to have in the main paper.

*Methods: It would be useful to be explicit in the main methods section about what was assumed about the % of notified individuals following app advice (eg effectively isolating) as this is key information. As a note, there is more that could be referenced in the supplementary about the % of contacts isolating (though perhaps not specifically as requested by the app), eg the Covid Social Study (Fancourt et al) and the CORSAIR study. I agree the ONS estimates are quite high compared to other surveys.

*Methods: In general, I would have preferred more of the detail from sections 1.9 and 1.10 in the supplementary materials to be in the main paper methods.

3. Consideration of costs - would it be possible for this analysis to also include assessment of the proportion of uninfected individuals notified to isolate? While extending the period over which contacts made were notified resulted in more contacts being notified and potentially more cases averted, the costs of this approach is in isolating more people who were not infected. Have the authors considered an assessment of the specificity of the app over time? It is possible that this would be a large piece of work in addition to the analyses already reported, but it would be useful to at least bring this up in the Discussion. (Although, removing more individuals from making non-household contacts could also prevent them from becoming infected, so it could be argued perhaps that their isolation is not entirely redundant.)

4. Please clarify in the paper Methods whether app users consent to their data being used for research purposes and if the research protocol was reviewed by an ethics committee, and if not, why this was not required.

Minor comments

1. The term 'risky social contacts' is used quite a few times and is unclear- please could this be defined. Are these non-household contacts? Or contacts made while a an individual is infectious and not notified? Or any contact made?

2. p1 When talking about reducing the 'size of the wave' please clarify whether this was number of infections, or height of the peak

3. Figure 1: please clarify that the denominator was the number of 16+ years aged individuals in each LTLA (not the total population)?

4. Figure 3A) please clarify whether these were all tests or PCR tests only, and if all tests, how LFD + confirmatory PCR test regimes were dealt with? Could these be identified as a single infection for one individual?

5. With regards to policy for testing contacts, I recall that contacts identified by NHS Test and Trace were allowed to take a PCR test even if they did not have symptoms from around the end of March 2021, not Aug 2021 as reported in the paper (without changing advice to isolate). This was a change to policy that previously did not allow free PCR testing to symptomless contacts. I checked archived gov.uk web pages on obtaining a free government PCR test and this does seem to have changed in late March 2021 (although I am not sure about the extent to which contacts were deliberately made aware of this and encouraged to do so).

6. Results p4 read: "Figure 4a shows the number of exposure notifications over time (blue), and demonstrates how this broadly follows the number of app-recorded cases (orange). The number of exposure notifications per app-recorded case (Figure 4b, red) is affected by population contact rates and by the proportion of users who consent to contact tracing after recording their positive result, which has varied over time but we estimate to be between 40 and 55% (Supplementary Figure S1)."

Presumably this would also reflect the probability of contacts being between app users? I think this point is partly made a couple of sentences later, but it might be worth being clear that this is to do both with the overall proportion of contacts being made by app users and the assortativity in app-using status.

7. Results p4: please could the authors clarify whether BOTH app-using parties need to have specifically consented to contact tracing for a notification to be made or only the user reporting a positive test?

8. Figure 4B : I think it would be helpful to separate out the CTAS contacts by household and non-household so that any differences in the trend relative to app notified contacts could be discerned (in addition to showing CTAS contacts overall).

9. Figure 6: Please clarify whether it is assumed that notified contacts are within the same LTLA as the app-reported cases that notified them. What is assumed about onwards transmission chains?

10. Figure 6 D and E: Do the authors have any interpretation as to the geographic patterns observed in the proportion of cases averted? These were not immediately clear to me based on the map describing the % uptake of the app and what I know about the geographical distribution of cases. Eg Why do the authors think this estimate was quite high in Cornwall?

11. p11: please clarify whether tests reported through the app were symptomatic PCR tests, other pCR tests or LFDs or all of these? Do the authors expect any biases in the types of tests reported through the app (either via type of tests ordered or type of tests otherwise reported)?

12. Discussion third sentence - please clarify how the app was 'aligned with local health policy' - this reads to me like the app was adapted according to specific needs, eg identified by local public health teams - was this the case?

13. How did the app compare to uptake in other settings once accounting for the % of app users who consent to contact tracing? Was this second step required in other settings also?

Responses to reviewers

We thank the reviewers for their consideration of the paper and helpful suggestions. Comments from reviewers are formatted in black font without italics; our responses are formatted in blue with italics.

We have edited the manuscript according to the Nature Communications formatting instructions. We have also made a further update which does not directly relate to any of the points raised in review: on 17 November 2022, UKHSA began publishing more detailed app data at a new URL <https://www.gov.uk/government/publications/nhs-covid-19-app-statistics> and so we have updated some of our text and references to incorporate this.

REVIEWER COMMENTS

Reviewer #1 (Remarks to the Author):

The paper uses publicly available data from the NHS about the use of its exposure notification app to report on the epidemiological impact of the app. More specifically, it tracked the app reported cases, whether notified users ended up submitting a positive test and finally it gave an estimate of the number of averted cases, hospitalizations and deaths.

I believe that the paper provides an important data point on the impact of digital contact tracing. However, I have struggled with identifying its novelty. I think the longitudinal aspect of the study is indeed interesting, but the findings are not novel. The fact that digital contact tracing can be/was effective has been established by a number of studies, including a couple by the authors. Hence, this paper is not necessarily adding significantly to the existing art.

The period with excessive notifications is quite interesting as such bugs can negatively impact the public perception of the app. Hence, it would have been really interesting to look into how that bug affected the adoption and use of the app.

The analysis of TPEAN is one of the most novel parts of the paper. Unfortunately it wasn't developed further. TPEAN varied over time. The authors rightly attributed that to changing vaccination levels, viral variants, policy, and behavior. Nevertheless, the paper did not attempt to find causal links between these factors, and the observed variability.

While I acknowledge the difficulty and limitations imposed by the aggregated and anonymized nature of the dataset, I believe that augmenting it with other data sources like

surveys and data on manual contact tracing or even anonymized app data from other countries could have helped improve the value of this work.

Our thanks to Reviewer 1 for these comments. We have two clarifications. Firstly, the data used is not all publicly available and we believe that the novelty of the paper includes not only longitudinal TPAEN and the estimates of cases, hospitalisations and deaths averted, but also the first comparison to manual tracing (Figure 4b) and our detailed assessment of the “accuracy” of the app (Figure 5). Secondly, the period with “excessive” notifications was not caused by a bug, but simply by the combination of the high number of cases, high population mixing, and high uptake of the app (Figures 1a, 3 and 4). Indeed, the accuracy of the app was particularly high in the months directly preceding that period (Figure 5d, e, f), and we have now emphasised this in the penultimate paragraph of the Results section together with discussion of the possible causes of the decreases in these measures in June-July 2021. We agree that causal links between our longitudinal analysis of TPAEN and other dynamic factors would indeed have been interesting but are sadly impossible given the data limitations you mention. App data sharing agreements are extremely stringent and while working with this data we are not allowed access to any other private data in UKHSA (e.g. to make a more detailed comparison to manual contact tracing data). Similarly, we do not have access to other countries’ private app data.

Reviewer #2 (Remarks to the Author):

In this paper the authors present an assessment of the epidemiological impact of the NHS COVID19 app in England and Wales, one year after its deployment, and can be seen as a follow up work after [Wymant et al, Nature 2021] that accounted for initial impact of the app 3 months after deployment. This follow-up paper is based on the former to quantify the impact of the app but adds a range of new insights, including (i) impact of a modification in the app settings, (ii) longitudinal analysis and effect of different variants, variation of uptake based on the app being needed to produce necessary QRs, user fatigue, vaccine roll out, etc. The main results depicted in the paper include (I) observing a clear signal of the app sensibility (users who were recently notified and get tested are more likely to get a positive test than those who were not recently notified) and (II) an assessment of the impact of the app in terms of transmission aversion (about 1 million cases, translating into circa 44k averted hospitalisations and circa 10k deaths). This, together with the fact that app deployment is a rather cheap public health intervention, confirms its usefulness for pandemic response.

The paper is clearly written, the methodology is well documented and the results very impactful. While this is indeed a follow up work that confirms early results by the same group, I believe it is still a much needed piece of work. The reason is because the data and analysis coming up from the NHS app deal with a scenario where the app uptake is relatively high –this being a clear bottleneck both in theory and in practice, as documented

by both modelling and experiments in recent literature--. It seems the NHS case can provide evidence of how a digital contact tracing technology could effectively contribute if (and this is a big if) adoption (user uptake) is sufficiently high and stable over time.

Accordingly, in general I recommend publication of this work in Nature Communications. Below I summarise a list of aspects which I believe could benefit from a more detailed explanation or consideration. The comments are not sorted in order of importance.

We thank the reviewer for their overall positive evaluation.

- The national uptake is given in terms of millions, might be a good idea to also give it in terms of percentage of the whole population so that results can be used for extrapolation and comparison with other works?

We agree that this is a useful addition for international comparison and have added a few comparative statistics in the first two paragraphs of the results section. These national values supplement the existing Figure 1b which shows the sub-national geographical variability.

- Comments on Fig1:

Panel A of Fig 1 displays the (geographically-aggregated) number of "active apps" throughout time, whereas panel B of the same figure depicts a geographically disaggregated yet time-aggregated visualization. I have several comments and questions on this Figure.

*** How does the data collection and its granularity allow you to determine that the app is actually "active" wrt, say, a situation where the user downloaded the app but then uninstall it or switches off the exposure notification system? I know that some explanation is reported in the Methods Section, but I think it would be helpful to clarify how this data was collected and to which extent this was compatible with the rather strict privacy-preserving paradigm. Note this type of data was not available "by design" in other contact tracing apps.**

We define the number of active users as the count of daily analytics packets received by the central server. These packets are sent only if the app is installed (they stop if it is uninstalled) AND the device has access to the internet, i.e. switched on and with a data connection. Switching the bluetooth exposure notification system on and off does not affect this measure. The GAEN protocol permits this collection of anonymous daily packets for the sake of monitoring proper functioning of the app. The app was approved by Google, Apple and the UK Information Commissioner's Office. We have moved some details about this from the Supplementary Material to the main Methods section and added clarifying details. In addition, we have now included "contact tracing enabled" users in our results, thanks to a new data release from UKHSA.

*** At a first order approximation, one could argue that population density might be roughly correlated with the R number (e.g. more populated regions might tend to yield more densely connected social contact networks, or might be prone to host super spreading events, etc). If true, this assumption would imply that from a transmission viewpoint it would be more critical to have a larger app adoption/adherence in those more densely population regions. I wonder if the authors have considered or taken into account this aspect in their estimation analysis. Spin-off analysis could also include checking whether there is a correlation between more densely populated regions and a higher adoption (visually it seems to be the case from the panel b?), correlation between app adoption (% population) and ICU density, and many others.**

A quick comparison (below) indicates that there is not a strong correlation between population density and app uptake at the LTLA level. We have checked the data for any natural experiments which could be exploited for further analysis, but unfortunately the variation in app uptake over time was quite uniform across LTLAs.

Our calculations of cases averted as a result of app notifications are performed separately for each LTLA, then summed to find a national total. In doing so, we are allowing for geographical variation in app use (specifically, the number of notifications received) and connecting this with

local case rates and R, which may be correlated with population density. Please see also our response to Reviewer #3 question 9 regarding a limitation of this approach.

It would be interesting to explore further the network effects of virus transmission and app use. Unfortunately for our analysis, the only “location” data we have on app users is limited to their self-declared postcode district (and, in some cases, Lower Tier Local Authority) at the point of app install. During times of strict social restrictions it might be more justifiable to make assumptions about where people are making their contacts, but in other analyses we have found that such assumptions quickly break down outside of these time periods. For example, we cannot distinguish between hypothetical users A and B who both register a low population density postcode district on install, but user A spends most of their time in that district whereas user B commutes daily to a densely populated region or workplace.

It is certainly the case that virus transmission occurs on social networks, and that a complete understanding of the app’s effect requires an understanding of the homogeneity of app use across a social network. This can be explored theoretically for example in agent-based models such as the Open-ABM (<https://journals.plos.org/ploscompbiol/article?id=10.1371/journal.pcbi.1009146>). High app use amongst highly connected individuals is certainly one driver of app effectiveness. However, the limited data collected by the NHS COVID-19 app means that it is not possible for us to analyse this any further in practice.

- Comments on Fig 2 and 3:

Results seem a priori impressive, where the percentage of positive tests associated to app users that had been notified increased up to 70%. This is, however, just a side of the story, and I am not sure the authors have presented this story in the most balanced and clear way. I'd suggest, if possible, to estimate some proxy of a confusion matrix, that would provide not only true positive rates but also false positive rates, etc. It is expected that the false positive rate is large (somehow related to the pingdemic concept), and the authors should discuss this aspect (for instance, they mention the number of cases detected increased after a change in the app settings, does this translate in a stable or increased false positive rate?) I think it is only fair to provide all these aspects of the data.

The measure that increases up to ~70% in Figure 3b is the proportion of national positive tests that are reported through the app; it does not incorporate notification status. We do not have data on true versus false positive rates of testing. Perhaps the reviewer is referring to Figure 5? With regards to the probability of testing positive shortly after notification, and the influence of the change in app settings in December 2020, it is possible to argue from Figures 5d, e and f that the measures in those figures decreased slightly following the change but quickly recovered - it does not seem to be a strong effect. The “accuracy” of the app in notifying infected users was high in the period preceding the pingdemic, and we have emphasised this in the penultimate paragraph of the Results section.

We define a proxy for the app's true positive rate through the quantity "TPAEN". As explained in the paper, this is the proportion of notified app users that report a positive test in a time window from notification to 14 days after quarantine, and it is an underestimate of the true fraction of notified users that are infected by a factor of 1.3 to 3 (ONS infection to case ratio) plus the "leaky pipeline" of notified users who do not take a test in a timely manner, or do not report their positive result.

- Abstract and Figs 5

One of the main results is a positive association between being notified by the app and testing positive, as compared to those not notified by the app. It is not clear to me what the control group is: is it a group of people that decided to get tested (because of whatever reasons but excluding app notification)? Or a random selection of people that were tested (like a random sampling)? I think this is important to clarify because the conclusions are different in both cases. From the abstract I had the impression it's the former, but from the text I then thought it's the latter.

If it's the latter, the results are nice but somehow expected. If it's the former, the results are much stronger because of the following (I'm slightly reaching, just for the sake of argument): one could argue that those citizens that adopt the app might be more sensitised to the overall pandemic situation, and that in turn might include (i) take lesser risks, (ii) "follow the rules" more strictly, and so on. Conversely, those users not engaged with the app might tend to be more relaxed. In that scenario, those citizens not using the app and yet voluntarily deciding to get tested would be more likely to get a positive test, if only because these more relaxed citizens might only go through the burden of getting tested when they have a clear case (having clear symptoms, having a positive case in the household, etc). Under this reasoning the result of the authors would be even stronger. Is it possible that the authors address both scenarios in panels of Fig 5?

A clarification would be needed.

We compare to two different control groups available to us: first (Figure 5d) we compare notified app users to random sampling of the population from the ONS survey, and second (Figures 5e and 5f) we compare notified app users to non-notified app users. In the Abstract we refer only to the second comparison (Figures 5e and 5f) for brevity, and we have tweaked the wording there and in the Results section to make this clearer. We agree that "a group of people who decided to get tested but not because they had an app notification" would be a nice control group to measure against, but unfortunately we do not have such data.

- Transmission mechanism and rationale for DCT.

In the early times of Covid digital contact tracing, when it was tacitly assumed that droplets were the main transmission mechanism, it seemed sensible to impose a 1.5 metre distance + some time window 15 min) as the threshold to consider a potential contact was established. Actually, these parameters were really unknown, and the truth was that the

1.5 metre part was more a criterion based on bluetooth quality (beyond 2m the signal intensity decay is not reliably correlated with actual distance). Even worse, with the realization that the virus has an airborne transmission, the 1.5 metre criterion seemed more arbitrary (or just based on technological considerations). In a nutshell, the whole criterion to determine when a contact has been established is, I believe, a very complicated issue with not obvious response from a infectious disease point of view. I think this aspect should be clearly acknowledged and discussed in the paper, as throughout this work (and surely, the rest of the work in DCT) assumes that these criteria are biologically-informed, which is far from clear to be the case. If new evidence has come to light and what I am saying is not true, I would be delighted, either way, I think this merits a discussion, because it is a clear limitation (or a premise of the usefulness of) the whole DCT paradigm. In this sense, the main part of the paper mentions a change in the detection criteria of the app but (not sure if I missed this) no detailed information of what the change is, and why is it sensible, is provided.

For the NHS COVID-19 app, users are notified if the risk score for at least one of the exposure windows (each lasting 30 minutes, set by the Google Apple Exposure Notification system as the unit of exposure) exceeds a given risk threshold. The risk scoring is based on a continuous function of distance in metres ($1/\text{distance}^2$ above 1m, 1 below 1m) multiplied by the duration of exposure within the 30-minute window, and by a two-level coefficient that accounts for infectiousness varying with respect to time of onset of symptoms. Therefore, there is actually no strict threshold on time or distance individually.

The app is calibrated such that the risk threshold is close to the 2m 15min threshold chosen by the NHS. (The risk threshold was actually higher but was changed twice in autumn 2020.) But at least in theory, the app can notify individuals based on less than 2 minutes of (very close) exposure, or at more than 4 meters' distance (but for 30 minutes).

The reviewer is right that all results presented in this paper depend on the specific choice of risk scoring and threshold, and we added a clarification about this in the text when discussing Figure 4a in the Results section.

Unfortunately the choice of both algorithm and threshold is not a choice that can be easily discussed in any detail in the paper, and any quantitative discussion requires a different set of app data compared to the ones used here. We will discuss risk scores in greater detail in a separate analysis in preparation.

Signed: Lucas Lacasa

Reviewer #3 (Remarks to the Author):

Summary

This paper reports on the NHS COVID-19 app, whose primary function was to notify contacts of recently diagnosed individuals with SARS-CoV-2 infection among app users and advise them to take the action recommended/required at the time (ie, quarantine at home, later, take a PCR test). The paper first describes the app's uptake and engagement, average numbers of notifications, and positive tests among notified users over time reflecting epidemiological dynamics and changing policies on social mixing. The authors then provide an estimation of the number of cases, hospitalisations and deaths likely averted through the app's contact tracing action, based on a number of assumptions, and conduct further sensitivity analyses to vary some of these assumptions.

Overall comments

Thank you for the opportunity to review this paper, which I found very interesting and informative. It is of high interest to the readership of this journal. Descriptions of the performance of COVID-19 tracing apps are important to conduct and report and this paper is thorough and thoughtful in doing so. I have a set of concerns about methods for attributing averted infections due to the app, some comments about report structure and about additional considerations. Apart from a few points about what detail is included where (below), I found the paper well-written and the figures were clear and nicely presented.

We thank the reviewer for their overall positive evaluation.

Major comments

1. Household versus non-household contacts

My main concern about the estimation of cases, hospitalisations and deaths averted due to the app, is around how household versus non-household contacts are treated. The transmission probability to household contacts is relatively high compared to other contact types, and it seems likely that two app users within a household are likely to be notified, given the probability of close, long-duration and repeated proximity. Household versus non-household contacts are not mentioned much in the paper, but I think there are some important differences between household and non-household contacts that need to be accounted for in assessing the impact of contact tracing. Household contacts are different in that they 1) are not protected from infection by isolation of the infected individual at home (or not as much as non-household contacts) and because 2) tracing

household contacts, whether manually or with the app, could be considered to be redundant, for many or most households. The second issue can be dealt with by varying the proportion of case notifications due to the app (as varied in the sensitivity analysis), but I am not sure how the first issue is dealt with in this analysis. The proportion of contacts made during the infectious period who are household contacts, and the proportion of transmission among household versus non-household contacts would also be expected to change over the course of the epidemic as policies around social mixing changed. Additionally, contacts of within-household cases were for most of this time period meant to begin isolation immediately upon the household index case's symptoms, so if following policy and depending on test delays, they would not have had out-of-household contacts during their infectious period, regardless of whether they are traced.

Presumably, there is not a definite way of identifying when notifications from the app are between household members - could this be clarified? I wasn't quite clear from the supplementary whether data was retained on the duration/repetition of contact which might help to indicate household versus non-household contacts, or, if collected, whether this data was made available for analysis.

I do think the paper needs an explanation for how household versus non-household contacts are dealt with in assessing the impact of the app on cases, hospitalisations and deaths, and perhaps some reconsideration of the results reported as the primary analysis as non-protection of household contacts by isolation of infected individuals (whether identified as such or not) should reduce the proportion of cases that the app could prevent.

I do agree with the point that the app should have been in a better position to identify potentially infectious out-of-household anonymous contacts relative to manual tracing.

Indeed, there is no reliable way of identifying household versus non-household contacts. A different source of data recorded by the app provides some information on contact durations which might be used as a proxy for household status, and we will be exploring this in detail in an upcoming paper. We agree that "The proportion of contacts made during the infectious period who are household contacts, and the proportion of transmission among household versus non-household contacts would also be expected to change over the course of the epidemic as policies around social mixing changed." We deal with the fact that household contacts are not protected from infection by isolation of the individual at home by treating "isolation" as a proportional reduction to one's "risky contacts". We note that we had not carefully defined this term previously, and have now clarified it along with the following reasoning in the Results section. We reason that very few app users are able to "perfectly" isolate, and suppose instead that an app notification resulted in an average reduction in risky contacts of 60% in the pre-Alpha and Alpha waves (lower 38%, upper 82%) and of 40% in the Delta wave (20%, 60%).

In our sensitivity analysis we further consider different levels of reduction of risky contacts for other interventions.

Regarding one test-positive household member causing an app notification for another household member - yes indeed, in this situation an app notification is likely redundant for reducing infections, and the same is true for manually traced contacts. We note that at some time points, receiving such a notification was still useful for accessing self-isolation payments. This is incorporated in our analysis where we discuss (previously in the Supplementary Materials, now in the main Methods) the overlap between app notifications and an individual suspecting or discovering their infectious status by another means, including word of mouth tracing between household members. We accept that this is poorly informed by data and have tried to be conservative in our estimates by a) allowing some effect from other PHSMs to precede app notifications (visualised in Figure S4 as the orange and green lines being lower than the blue line before the app notification is received on day 5), b) attributing to an app notification a conservative reduction in risky contact rates, c) attributing no contribution to the app notification after the arrival of another PHSM, which includes word-of-mouth tracing. Of particular relevance here is (c): apparently “redundant” information may be interpreted as confirming or even increasing one’s risk of being infected and may increase adherence, but we have not been able to quantify this.

2. Level of detail in the main paper versus supplementary

I would have preferred more methodological detail in the main paper itself, not only in the supplement. Information that I would consider highly relevant to the extent to which I trusted findings was in the supplementary section but not in the Methods. In some cases, because the Results is prior to the methods section, I think it would be helpful to include brief sub-clauses defining key terms in the Results when these terms are first used.

Specific instances include:

***Results - could a very brief sub-clause defining ‘active user’ be included in the Results text itself (first paragraph)?**

***Results - ditto ‘app-reported cases’**

***Methods- further clarity about the time-varying parameters underlying the estimation of averted cases, hospitalisations and deaths (eg reflecting different test result return delays), would be helpful to have in the main paper.**

***Methods: It would be useful to be explicit in the main methods section about what was assumed about the % of notified individuals following app advice (eg effectively isolating) as this is key information. As a note, there is more that could be referenced in the supplementary about the % of contacts isolating (though perhaps not specifically as requested by the app), eg the Covid Social Study (Fancourt et al) and the CORSAIR study. I agree the ONS estimates are quite high compared to other surveys.**

***Methods: In general, I would have preferred more of the detail from sections 1.9 and 1.10 in the supplementary materials to be in the main paper methods.**

We agree that it is preferable to include more methodological details in the main paper and, upon realising that we are well within the word count, we have moved some substantial detail from the Supplementary Material to the main Results and Methods sections. In particular, we moved details of the TPAEN calculation, and:

- *A brief definition of active users added to the first paragraph of the Results*
- *A brief definition of app-reported cases added to the third paragraph of the Results*
- *We have been more explicit about the fact that timings vary between individuals and over calendar time, for both app notifications and other interventions*
- *We have moved to the main Methods our estimates of the reduction in risky contacts as a result of app notification (part of section 1.9)*
- *We have provided more detail in the main Methods about the sensitivity analysis (from section 1.10)*
- *We were unfortunately unable to find any statistics from the Fancourt et al Covid Social Study relating to contact tracing and adherence. We found a result about overall compliance with guidelines decreasing during our period of study, which we have added as a reference to support our assumption in the Supplementary Materials about adherence during the Delta wave. Similarly, as far as we can tell, the CORSAIR study covers adherence to self-isolation if symptoms are present and “intention to share details of close contacts”, but unfortunately nothing to inform our work on adherence to contact tracing alerts.*

3. Consideration of costs - would it be possible for this analysis to also include assessment of the proportion of uninfected individuals notified to isolate? While extending the period over which contacts made were notified resulted in more contacts being notified and potentially more cases averted, the costs of this approach is in isolating more people who were not infected. Have the authors considered an assessment of the specificity of the app over time? It is possible that this would be a large piece of work in addition to the analyses already reported, but it would be useful to at least bring this up in the Discussion. (Although, removing more individuals from making non-household contacts could also prevent them from becoming infected, so it could be argued perhaps that their isolation is not entirely redundant.)

We have attempted to perform cost-benefit analyses but have found that anything beyond the most simple heuristic measures run into challenges, which include:

- *The limitations of our estimates - the difficulty of defining the counterfactual (e.g. cases averted by the app reduce the quarantine those cases would have caused through non-app interventions), the fact that our estimates are likely to be underestimates, and the important one you note: the effect we refer to as “the network effects of app usage” which sees notifications actually preventing infections as well as preventing transmissions. Please see also our comments on the false positive rate in response to Reviewer 2.*

- The “cost” of isolating would have varied throughout the period of study, for example in early 2021 when schools and most workplaces were closed, being asked to isolate was likely to be less costly to the average user than later in 2021 when there was more freedom of movement. We do not have data on the number of app users who were eligible for self-isolation payments. Finally, we do not know adherence rates and there was never a legal requirement to isolate as a result of an app notification.
- The separation of (costs of) interventions. During the period of study, contact tracing in the NHS COVID-19 app could only be triggered by a positive PCR result. This relied on a complex infrastructure of testing within NHS Test and Trace (later UKHSA).

4. Please clarify in the paper Methods whether app users consent to their data being used for research purposes and if the research protocol was reviewed by an ethics committee, and if not, why this was not required.

Ethical approval was not required because our analysis was performed on routinely collected, anonymised and aggregated data that cannot be traced back to individuals, from a database built with the primary purpose of supporting public health. Please see this link for more information <https://www.hra.nhs.uk/covid-19-research/guidance-using-patient-data/#research>. We have added this before the Declaration of interests.

Minor comments

1. The term ‘risky social contacts’ is used quite a few times and is unclear- please could this be defined. Are these non-household contacts? Or contacts made while a an individual is infectious and not notified? Or any contact made?

We have now defined what we mean by a “risky contact” in the penultimate paragraph of the Methods.

2. p1 When talking about reducing the ‘size of the wave’ please clarify whether this was number of infections, or height of the peak

We have clarified that we mean the number of cases.

3. Figure 1: please clarify that the denominator was the number of 16+ years aged individuals in each LTLA (not the total population)?

We have clarified that this is total population - we were unable to find the relevant age data at the LTLA level.

4. Figure 3A) please clarify whether these were all tests or PCR tests only, and if all tests, how LFD + confirmatory PCR test regimes were dealt with? Could these be identified as a single infection for one individual?

We have clarified in the Methods section that these are PCR and LFD tests. In that paragraph we note that “it is not possible to distinguish if a user enters positive tests over multiple days, and may therefore be counted repeatedly” and that this is more likely to distort the measure towards the end of the period of study as LFDs became more widely available, in contrast to the government dashboard data which is deduplicated by individual.

5. With regards to policy for testing contacts, I recall that contacts identified by NHS Test and Trace were allowed to take a PCR test even if they did not have symptoms from around the end of March 2021, not Aug 2021 as reported in the paper (without changing advice to isolate). This was a change to policy that previously did not allow free PCR testing to symptomless contacts. I checked archived gov.uk web pages on obtaining a free government PCR test and this does seem to have changed in late March 2021(although I am not sure about the extent to which contacts were deliberately made of aware of this and encouraged to do so).

We have corrected that sentence in the Results section and added detail.

6. Results p4 read: “Figure 4a shows the number of exposure notifications over time (blue), and demonstrates how this broadly follows the number of app-recorded cases (orange). The number of exposure notifications per app-recorded case (Figure 4b, red) is affected by population contact rates and by the proportion of users who consent to contact tracing after recording their positive result, which has varied over time but we estimate to be between 40 and 55% (Supplementary Figure S1).”

Presumably this would also reflect the probability of contacts being between app users? I think this point is partly made a couple of sentences later, but it might be worth being clear that this is to do both with the overall proportion of contacts being made by app users and the assortativity in app-using status.

That’s right - we have clarified this in paragraph 4 of the Results.

7. Results p4: please could the authors clarify whether BOTH app-using parties need to have specifically consented to contact tracing for a notification to be made or only the user reporting a positive test?

Indeed there are two parts to this which we have now clarified in paragraph 4 of the Results.

8. Figure 4B : I think it would be helpful to separate out the CTAS contacts by household and non-household so that any differences in the trend relative to app notified contacts could be discerned (in addition to showing CTAS contacts overall).

We agree that this is a helpful perspective and have added the breakdown of CTAS data by household and non-household contacts to Figure 4b.

9. Figure 6: Please clarify whether it is assumed that notified contacts are within the same LTLA as the app-reported cases that notified them. What is assumed about onwards transmission chains?

We do not make an assumption about whether notified contacts are within the same LTLA as the app-reported cases that notified them, but we do assume that the onwards transmission chain remains within the LTLA of the notified contact. We have now clarified this in Discussion.

10. Figure 6 D and E: Do the authors have any interpretation as to the geographic patterns observed in the proportion of cases averted? These were not immediately clear to me based on the map describing the % uptake of the app and what I know about the geographical distribution of cases. Eg Why do the authors think this estimate was quite high in Cornwall?

Uptake is one of many determinants of the app's effect. Other determinants include "engagement" (inputting positive tests, consenting to contact tracing), clustering of app users, and prevalence at times when the app was able to have greater impact, e.g. when test turnaround times were relatively short, cases were concentrated amongst app users (i.e. lower case rates among the under 16s and elderly) and when there were more risky contacts outside the home which had the potential to be reduced by an app notification. We have added a sentence about this at the end of the Results section.

11. p11: please clarify whether tests reported through the app were symptomatic PCR tests, other pCR tests or LFDs or all of these? Do the authors expect any biases in the types of tests reported through the app (either via type of tests ordered or type of tests otherwise reported)?

We have clarified in the Methods section that the tests reported through the app were both PCR and LFDs. We do not have any data on why the PCR tests were taken, i.e. whether the individual was eligible due to symptoms or other reasons, and we are unaware of any biases surrounding test type. Our estimates of cases averted are based only on positive test results, and we generally expect false positives to be rare.

12. Discussion third sentence - please clarify how the app was 'aligned with local health policy' - this reads to me like the app was adapted according to specific needs, eg identified by local public health teams - was this the case?

We have added some sentences to clarify that the app was updated each time that legal requirements and/or public health advice changed, and that these varied by location.

13. How did the app compare to uptake in other settings once accounting for the % of app users who consent to contact tracing? Was this second step required in other settings also?

To the best of our knowledge this data is unfortunately not available for other comparable apps.

REVIEWERS' COMMENTS

Reviewer #2 (Remarks to the Author):

The authors have adequately addressed my comments. I support publication of this manuscript in its current form.

Reviewer #3 (Remarks to the Author):

Second review (after first set of revisions):

The edits that the authors have made to improve clarity of the main article text, pulling more information from the supplementary materials, have been very helpful.

I appreciate the explanation given in response to my concerns as to how household contacts are considered, and the explicit mention of this on p10 as part of the uncertainty in the effects of a contact notification.

I accept that an analysis of 'costs' of digital contact tracing (eg requiring uninfected contacts to isolate) would be difficult to do given the arguments made, and probably any attempt to do so would constitute a separate piece of work.

My other minor concerns have been mentioned as limitations or addressed.

Liz Fearon